# Reliability and Validity Analysis of the Korean Version of the Affinity for Technology Interaction Scale

**DOI:** 10.3390/healthcare11131951

**Published:** 2023-07-06

**Authors:** Taehui Kim, Seyeon Park, Miri Jeong

**Affiliations:** 1Department of Nursing, Joongbu University, Geumsan-gun 32713, Republic of Korea; 2College of Nursing, Chungnam National University, Munhwa-ro 266, Daejeon 35015, Republic of Korea

**Keywords:** affinity for technology interaction, nurse, reliability, scale, validity

## Abstract

This study aimed to translate the affinity for technology interaction (ATI) scale into Korean and examine its validity and reliability to measure nurses’ interactions and affinity with digital healthcare. Data from 154 nurses employed by general hospitals were analyzed. This developmental and psychometrical evaluation of the ATI scale included a translation, a pilot test, and psychometric properties. Concurrent validity, content validity, construct validity, and reliability testing were completed. The corrected item–total correlation was below the standard of 0.3, and the content validity index was >0.8. The Kaiser–Meyer–Olkin and Bartlett sphericity test values were 0.81 and *Χ*^2^ = 496.25 (*p* < 0.001), respectively. The exploratory factor analysis (EFA) result was extracted as two factors, with an overall variance of 60.52%. The correlation between the ATI scale and the Quality Improvement Information System acceptance tool was examined to confirm concurrent validity and showed a significant correlation. Cronbach’s α was 0.75, indicating adequate reliability. ATI’s internal, construct, and concurrent validity demonstrated its suitability as a tool. Therefore, assessing nurses’ information and communication technology proficiency and developing strategies for boosting it would be possible.

## 1. Introduction

Robotics, the Internet of Things (IoT), artificial intelligence (AI), and big data technologies were expected to significantly alter the healthcare industry and advance digital healthcare and medical technology [1,2].

The convergence of advanced technology and medical devices in clinical centers has resulted in more accurate diagnoses and better treatment [3]. In addition, big data collected through newly developed medical devices and AI are expected to facilitate accurate diagnoses and improve the treatment of diseases [3]. Therefore, the development of digital healthcare enables efficient patient-specific care and management of diagnosis, prevention, and treatment of disease and rehabilitation. In line with these changes in the medical workplace, there is an increasing need for healthcare workers to provide digital healthcare [4].

Nurses at medical facilities in Korea must be proficient in safety management, infection control, medical device management, and direct care, such as checking vital signs and providing medication and injections [5,6]. Medical device management (i.e., indirect nursing care) must be handled accurately and quickly for effective patient care and treatment. Various medical devices are used in medical workplaces and are being continuously upgraded with the development of technology. Therefore, nurses are regularly trained to use these medical devices. The development of technology and new machines can reduce nurses’ tasks, but nurses have to deal with the psychological burden of learning to operate them [7]. Nurses’ awareness and understanding of digital healthcare are lower than those of other health workers’ [8]. One study found that the telemedicine awareness of nursing students and the general public was more positive than that of nurses in Korea [9]. Thus, nurses’ awareness of digital healthcare needs to be increased. To this end, customized education can be conducted if the degree of nurses’ affinity for the new system is identified. This affinity will increase nurses’ ability to adapt to the changing medical technology, thus making it possible to respond to technological changes quickly. Furthermore, nurses may face situations where they must rapidly assess a patient’s condition and make critical decisions in the patients’ best interest. This assessment demands the ability to comprehend and analyze clinical information. In addition, a previous study described that user coping and adaptation to a new system is associated with knowledge of the system, a degree of intimacy, a general ability to interact with the technology system, individual disposition, and personality [10]. However, no scale measures nurses’ adaptation, coping mechanisms, and affinity for new technologies.

The affinity for technology interaction (ATI) scale developed in Germany reflects these characteristics. Two factors are essential to cope with a new system successfully: (i) skill for and knowledge of a particular system and (ii) personality characteristics [10,11]. The ATI scale was developed based on users’ interaction style with the system—whether users actively interact or avoid interaction with the technical system [10,11]. Generally, users with high ATI scores explore new technologies and deal with problems and malfunctions of systems in use, whereas users with low ATI scores are more likely to panic [12]. Franke et al. viewed continuous adaptation to regulation and problem-solving using new technical systems as an immediate and avoidant approach [10,12]. ATI is not merely a simple evaluation of technological knowledge; it also grasps individual tendencies [12,13]. It can provide education tailored to the characteristics of participants who use new technology. Therefore, such measurement can help distinguish between nurses with high and low ATI scores. The capacity of nurses with low ATI scores to adjust to new systems can be strengthened by imparting the necessary education.

The ATI scale was developed using nine items considering the importance of user-centered design and human–computer interactions. Depending on the demographic characteristics of participants, the reliability of the original ATI scale varied, but Cronbach’s α ranged from 0.83 to 0.94. The validity and distribution of ATI score values were satisfactory [11]. It has been translated and validated in English, Italian, Spanish, Romanian, Dutch, Persian, and German (available from 2021.12.15 ATI scale, ati-scale.org). The Korean translation of the ATI scale will help identify nurse characteristics related to the new technology. This information will provide a foundation for delivering customized education to nurses. Therefore, this study aims to translate the ATI scale into Korean and verify its validity and reliability to objectively measure the interaction and affinity of nurses toward digital healthcare.

## 2. Materials and Methods

### 2.1. Participants

The study participants were nurses who worked in general hospitals and dealt with various medical devices and technologies. Nurses who worked at local hospitals were excluded due to the substantive difference in technology used in local and general hospitals. Home-visiting nurses were also excluded. The type of ward or work was not a criterion for exclusion. The study participants had more than one year of experience working in general hospitals with more than 300 beds.

Those who understood the purpose and method of the study and consented to participate were selected. Data were collected through an online survey from 7 June to 27 August 2021. An exploratory factor analysis (EFA) requires 5–10 samples per question to confirm construct validity [14]; our scale contained 9 items, requiring a minimum of 90 participants. Data were collected from 155 nurses, and 1 participant was excluded from the analysis owing to insincere responses. Therefore, the number of participants included in the final analysis was 154.

### 2.2. Measures

#### 2.2.1. Affinity for Technology Interaction Scale

The ATI scale consists of nine items scored on a 6-point Likert scale. The responses are coded as follows: 1 = *completely disagree*, 2 = *largely disagree*, 3 = *slightly disagree*, 4 = *slightly agree*, 5 = *largely agree*, and 6 = *completely agree* [11]. Items 3, 6, and 8 are worded negatively. The Cronbach’s ⍺ of ATI in a previous study was 0.83–0.92 [10]. In this study, it was 0.794.

#### 2.2.2. Quality Improvement Information System Acceptance Measures

The Quality Improvement Information System (QIIS) acceptance measures assesses the acceptance of the quality improvement of an information system among long-term care workers. It consists of 16 items scored on a 5-point Likert scale: 1 = *completely disagree*, 2 = *disagree*, 3 = *moderate*, 4 =*agree*, and 5 = *completely agree*. The total score ranges from 16 to 80. It includes the five dimensions of usage intention, perceived usefulness, perceived ease of use, social influence, and innovative characteristics [15]. Cronbach′s ⍺ of this scale was 0.91.

#### 2.2.3. Numeric Rating Scale

Subjective ATI was measured using the numeric rating scale (NRS). The participants were instructed to indicate their ATI level on a line, ranging from 0 on the left to 10 on the right. A higher score indicated a greater ATI level.

### 2.3. Procedures

We employed a cross-sectional descriptive study design to develop and conduct a psychometrical evaluation of the ATI scale. This study used an ATI tool developed in Germany in 2019, which measures the degree of ability to understand and analyze clinical information. It consists of 9 questions [10]. First, the scale was obtained for use by the developer. Then, we undertook the following three phases: (1) translation and back translation, (2) pilot test, and (3) psychometric properties.

#### 2.3.1. Phase Ⅰ: Translation

We conducted translation and back translation for cross-cultural adaptation of the scale [16]. Three nursing professionals who were fluent in English translated the ATI scale into Korean. Then, the translations were synthesized, and a single questionnaire was obtained. Two experts back-translated the Korean version of the scale into English. No difference in meaning was found between the original and reverse translation scales. The Korean version was agreed on after discussion among the researchers and translators.

#### 2.3.2. Phase Ⅱ: Pilot Test

A pilot test was conducted to prevent any misinterpretation of meaning. It was conducted among ten clinical nurses who used the first version of the questionnaire. The technical system was described at the top of the questionnaire, as it is unfamiliar to Koreans. Both Korean and English versions were recorded for each item.

#### 2.3.3. Phase Ⅲ: Psychometric Properties

We performed content, construct, and concurrent validity tests. Cronbach’s ⍺ was used to evaluate the reliability of the ATI scale.

### 2.4. Analysis

The collected data were analyzed using SPSS WIN Version 24.0 (Chicago, IL, USA). A Kaiser–Meyer–Olkin (KMO) test was performed to check the validity and adequacy of the dataset by confirming the partial correlations between the variables [17], and a Bartlett sphericity test was performed to confirm the suitability of the data for factor analysis [18]. The mean and standard deviation were checked to confirm the clustering possibility and contribution of the items. Inter-item and corrected item–total item correlations were confirmed using Pearson’s correlation. As a result, a value with a correlation coefficient between items of 0.3 and higher was selected [19]. Factors with an eigenvalue of 1.0 or higher were extracted using the Direct Oblimin Rotation method [20]. Items with a factor loading of 0.4 or higher were extracted, and the factors to which these items belong were named. QIIS acceptance measures developed by Lee et al. (2017) were used to verify concurrent validity.

### 2.5. Ethical Considerations

This study was approved by the Institutional Review Board of Joongbu University (IRB No. 1040117-201311-HR-003-01). The agreement included the following: the purpose of the study, contents, procedures, and methods, freedom of participation, guarantee of anonymity, and assurance that the data collected would not be used for any purpose other than research. All participants who responded to the questionnaire consented and were compensated for their time.

## 3. Results

### 3.1. Demographic Characteristics

Most participants were female (95.5%), with an average age of 33.82 ± 5.99 years. Most worked in wards (40.9%), and others worked in operating rooms, emergency rooms, healthcare quality improvement departments, infection control departments, and administrative management departments (Table 1).

### 3.2. Distribution

The total mean score was 4.00 ± 0.79. The skewness of each item ranged from −0.48 to 0.35, and the kurtosis ranged from −0.63 to 0.83; hence, it did not violate the normal distribution (Table 2).

### 3.3. Validity

#### 3.3.1. Content Validity

Ten experts determined the content validity of the Korean version of the ATI scale. The content validity index was >0.8.

#### 3.3.2. Construct Validity

Item analysis and EFA were used to evaluate construct validity. The range of corrected item–total correlation was 0.36–0.61 for Factor 1 and 0.50 for Factor 2 (Table 3). No items were below the standard 0.3. Therefore, all nine questions were used for the EFA. The KMO and Bartlett sphericity test values were checked to verify the suitability of the EFA; the KMO value was 0.81, and the Bartlett sphericity test value was x^2^ = 496.25, *p* < 0.001. Therefore, the suitability of the EFA was confirmed. Factors with eigenvalues greater than or equal to 1.0 were identified using the maximum likelihood method, and two factors with eigenvalues greater than or equal to 1 were extracted. The elbow point was checked using a scree plot, and two factors were determined to be appropriate (Figure 1). Factor loading of 0.70 or higher was extracted as Factor 1; its eigenvalue was 3.92, and its variance was 43.55%. The factor loading of Factor 2 was 0.86; its eigenvalue was 1.53, and its variance was 16.94%. The overall variance of these two factors was 60.52% (Table 4).

#### 3.3.3. Concurrent Validity

The correlation between the ATI scale and the QIIS acceptance tool was confirmed as evidence of concurrent validity. A statistically significant relationship was observed between the ATI scale and the QIIS acceptance tool (*r* = 0.591; *p* < 0.001). NRS of ATI and ATI total showed a correlation (*r* = 0.459; *p* < 0.001), as did ATI Factor 2 and QIIS acceptance (Table 5).

### 3.4. Internal Consistency

Cronbach’s α was used to confirm the reliability of the Korean version of the ATI scale.

It was found to be 0.86 in Factor 1 and 0.66 in Factor 2. Cronbach’s α was 0.79, indicating reliability.

## 4. Discussion

This study verified the reliability and validity of the Korean version of the ATI tool by modifying and supplementing the ATI tool developed by Franke et al. [10] to objectively measure the digital familiarity of nurses in the Fourth Industrial Revolution. The Fourth Industrial Revolution has promoted digital health care, and the medical field is rapidly changing. Therefore, the target participants of the study were nurses. New nurses with less than one year of experience were excluded from the study because they were gradually adapting to clinical practice and experienced physical and mental burdens and anxiety [21]. Furthermore, technical and digital systems are different depending on the size of the hospital. This study targeted nurses working in general hospitals with more than 300 beds [22,23].

The KMO and Bartlett sphericity test values were checked to verify EFA; the KMO value was 0.81, and the Bartlett sphericity test value was x^2^ = 496.25, *p* < 0.001. These values were appropriate because KMO is greater than 0.6, and Bartlett’s sphericity test value is *p* < 0.05 [17,18]. EFA was conducted, and the result was extracted as two factors. One factor comprised questions 1, 2, 3, 4, 5, 7, and 9. Question 3 was reverse-scored. The other factor was composed of reverse-scored questions 6 and 8.

To deal with the technical system, ATI developed two relevant factors: knowledge/skill and personality facets. It also contains two perspectives, interest and intensive interaction, for successful technology interactions in the new system [11].

The presence of two factors in this study is not due to a difference in relevant factors or perspectives. Questions 6 and 8 are sentences that begin with “It is enough for me”. These are treated as reverse questions in the original scale. However, when the phrase “It is enough for me” is translated into Korean, its connotation in Korean culture differs from that in English-speaking cultures. In English, this phrase indicates contentment or satisfaction with what one has.

In Korean, it means, “Because I have been working hard for so long, this will do”, “I have done well so far; therefore, I do not have to try anymore”, or “I have worked hard up until now, so I do not need to put in any more effort”. Hence, when “It is enough for me to know the basic functions of a technical system” is translated into Korean, subjects seem to interpret it as “I have studied the basic functions of the technical system, so I know enough. I do not need to know more.” In addition, when translating English sentences into Korean, recognizing sentences that do not include “not” as those with negative meanings is challenging.

Furthermore, the subjects are nurses. Nurses have more substantial responsibilities and obligations than people in other occupations. They were thought to have understood the Korean meaning of “It is enough for me” as “I have worked hard so far, so I do not have to try anymore.” Therefore, these two questions are grouped into one.

The researcher in charge of translation should translate the questionnaire in consideration of the context being talked about instead of performing a literal translation [24]. Despite the translation and reverse translation, the differences in occupation, cultural characteristics, and cognition of the participants are thought to have shown in these results. Equivalence was not fully considered in the cross-cultural adaptation of questionnaires [25]. This finding should be confirmed through repeated studies involving the same and other groups in the future. However, further discussion is needed related to paraphrasing to suit Koreans’ emotions, cognitions, and perceptions to convey the intended meaning.

Inter-item and corrected item–total correlations were above 0.3 in construct validity [19]. The factor loading was greater than 0.7 for all items, and the eigenvalue was appropriate [20]. A golden standard Korean tool to measure concurrent validity for ATI precisely has not yet been developed. Affinity and acceptance of technology are critical and related factors [26]. NRS was used as a tool to measure one’s own ATI simply and intuitively, without understanding letters. The QIIS acceptance tool was developed based on the TAM (Technology Acceptance Model) to measure the nurses’ acceptance of hospital information systems [15]. A correlation analysis was performed using the QIIS acceptance tool, with an NRS added for concurrent validity. A correlation analysis confirmed that there was a relationship between technology affinity and QIIS. As a result, the ATI total score, NRS, and QIIS acceptance tool were significantly correlated, confirming concurrent validity.

Cronbach’s α was calculated to confirm the internal consistency. Cronbach’s α of the total questions was 0.794. For Factor 1, it was 0.865, and for Factor 2, it was 0.660. Cronbach’s α of the original scale was 0.83–0.92 [10]. In this study, a lower value was found than that in the original scale. Cronbach’s α of Factor 2 was low. However, the number of items in the scale affects Cronbach’s α. As the number of items increases, α increases [27]. In addition, an α of 0.65–0.80 is accepted as “adequate” in human dimensions research [27]. Therefore, K-ATI’s Cronbach’s α is adequate.

This study aimed to translate ATI into Korean and verify its validity through a factor analysis. The test results showed that the scale was valid and reliable. However, there were some limitations to this study. While the original scale had established validity through a large sample size, this study only involved a specific group of 154 nurses. The appropriate sample size suggestion varies among scholars [20]. Costello and Osborne suggested a sample size of 5–10 participants per item, while other scholars suggest a minimum sample size of 300 participants [20]. Therefore, the target population must be expanded, and a sample size of at least 300 must be secured to verify the scale’s validity.

The original scale consisted of a single dimension. However, this study divided nine items into two dimensions. The division may be due to the differences in linguistic perception among the subjects caused by the Korean translation. Therefore, further discussion is needed regarding whether to include the meaning of “not” in the translation to convey a negative sense.

Despite the limitations of this study, it is meaningful in that it verifies the validity of the Korean translation of a scale that can measure people’s resistance to or familiarity with new technologies or systems in the Fourth Industrial Revolution era. This scale can assess the level of adaptation to information processing and hospital systems necessary for nursing in a digitized medical environment. Additionally, it will be used to seek education and improvement strategies by measuring nurses’ understanding of the new system.

## 5. Conclusions

This study assessed the reliability and validity of ATI, a self-reported information and communication technology assessment tool, among nurses working at a Korean hospital. The nine-question Korean version of the ATI has adequate reliability and validity, making it a suitable tool for assessing Korean nurses’ knowledge regarding information and communication technologies. Future research can focus on re-evaluating the tool’s consistency through repeated research. Furthermore, we expect the K-ATI to be used as an objective tool to evaluate the ATI of nurses who experience continuously changing technologies and informatization.

## Figures and Tables

**Figure 1 healthcare-11-01951-f001:**
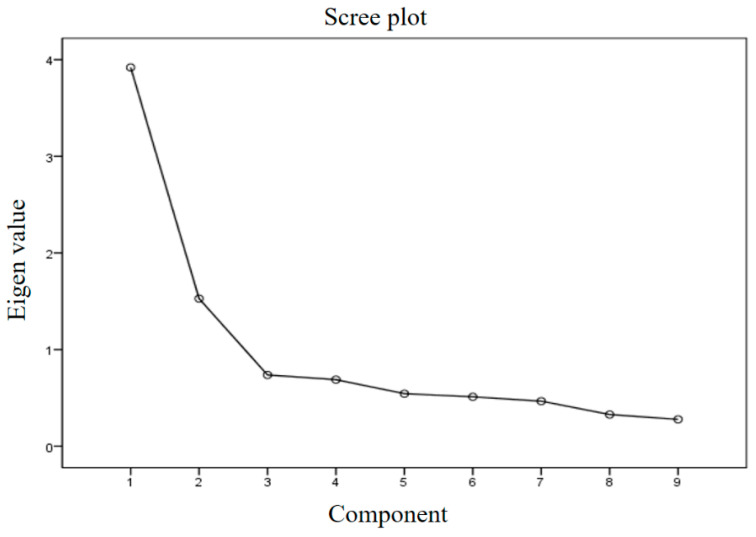
Scree plot.

**Table 1 healthcare-11-01951-t001:** Participant demographics (N = 154).

Variable	Category or Range	Mean ± SD or N(%)
Sex	Male	7 (4.5)
	Female	147 (95.5)
Age (years)	20s	41 (26.6)
	30s	87 (56.5)
	≥40s	26 (16.9)
	Range: 24–52	33.82 ± 5.99
Education	Associated degree	25 (16.2)
	Bachelor’s	107 (69.5)
	Master’s	22 (14.2)
Work unit	Ward	63 (40.9)
	Outpatient	17 (11.0)
	Intensive care unit	34 (22.1)
	Others	40 (26.0)
Position	Staff nurse	132 (85.7)
	Charge nurse	18 (11.7)
	Head nurse	4 (2.6)
Work type	Full-time work	64 (41.6)
	Shift work	90 (58.4)
Total clinical career	<5	22 (14.3)
(years)	5–< 10	55 (35.7)
	10–< 15	32 (20.8)
	15–< 20	32 (20.8)
	≧20	13 (8.4)
	Range: 1–30.1	11.13 ± 6.00

SD = Standard Deviation.

**Table 2 healthcare-11-01951-t002:** Item distribution.

Items	Mean ± SD	Skewness		Kurtosis	
		Statistics	S.E.	Statistics	S.E.
ATI1	4.26 ± 0.94	−0.21	0.195	−0.275	0.389
ATI2	3.86 ± 1.00	−0.20	0.195	0.340	0.389
ATI 3R	4.29 ± 0.81	0.21	0.195	−0.062	0.389
ATI4	4.12 ± 0.84	−0.17	0.195	−0.139	0.389
ATI5	4.26 ± 0.74	0.13	0.195	0.832	0.389
ATI6R	3.68 ± 1.09	0.23	0.195	−0.629	0.389
ATI7	4.04 ± 0.84	−0.48	0.195	−0.187	0.389
ATI8R	3.20 ± 0.88	0.35	0.195	−0.545	0.389
ATI9	4.27 ± 0.78	−0.10	0.195	0.171	0.389

SD = Standard Deviation; S.E. = Standard Error.

**Table 3 healthcare-11-01951-t003:** Exploratory factor analysis.

Items	Contents	Communalities	Factor 1	Factor 2
ATI4	When I have a new technical system in front of me, I try it out intensively.	0.67	0.82	0.06
ATI1	I like to occupy myself in great detail with technical systems.	0.60	0.78	0.03
ATI9	I try to make full use of the capacities of a technical system.	0.56	0.75	0.03
ATI2	I like testing the functions of new technical systems.	0.54	0.74	0.11
ATI5	I enjoy spending time becoming acquainted with a new technical system.	0.56	0.73	−0.08
ATI7	I try to understand how exactly a technical system works.	0.53	0.70	0.26
ATI3R	I predominantly deal with technical systems because I have to.	0.49	−0.70	0.00
ATI8R	It is enough for me to know the basic functions of a technical system.	0.75	−0.01	0.86
ATI6R	It is enough for me that a technical system works; I do not care how or why.	0.74	0.14	0.86
	Eigenvalue		3.92	1.53
	Variance (%)		43.55	16.97
	Cumulative variance (%)		43.55	60.52
	Range of corrected item–total correlation (*r*)		0.53–0.62	0.50
	Cronbach’s α, total: 0.79		0.86	0.66

ATI = Affinity for Technology Interaction.

**Table 4 healthcare-11-01951-t004:** Item correlation.

Items		Inter-Item Correlation							Corrected Item–Total Correlation
	ATI1	ATI2	ATI3R	ATI4	ATI5	ATI6R	ATI7	ATI8R	
ATI1	1								0.616
ATI2	0.612	1							0.595
ATI3R	0.501	0.396	1						0.529
ATI4	0.511	0.530	0.552	1					0.668
ATI5	0.466	0.383	0.463	0.630	1				0.538
ATI6R	0.030	0.139	0.062	0.107	0.047				0.229
ATI7	0.434	0.480	0.369	0.501	0.363	0.192	1		0.603
ATI8R	0.039	0.002	−0.010	0.010	−0.081	0.503	0.113	1	0.125
ATI9	0.517	0.437	0.382	0.514	0.500	0.115	0.560	−0.040	0.589

ATI = Affinity for Technology Interaction; R = Reverse.

**Table 5 healthcare-11-01951-t005:** Correlation between ATI_NRS and QIIS.

	ATI_Total	ATI_NRS	QIIS
ATI_total	1		
ATI_NRS	0.459 **	1	
QIIS	0.591 **	0.552 **	1

** *p* < 0.001. ATI = Affinity for Technology Interaction; NRS = Numeric Rating Scale; QIIS = Quality Improvement Information System.

## Data Availability

The data presented in this study are available upon reasonable request from the corresponding author. The data are not publicly available due to privacy and ethical restrictions.

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
