# Peer review of "Reliability and Validity Analysis of the Korean Version of the Affinity for Technology Interaction Scale"

_healthcare, 2023, doi:10.3390/healthcare11131951_

Round 1

Reviewer 1 Report (Previous Reviewer 1)

ID: healthcare-2457533

Title: Reliability and validity analysis of the Korean version of the Affinity for Technology Interaction scale

Thank you for providing a chance to review this manuscript.

Detailed information:

Abstract

Lines 19-21, page 1: Why was the full name used correctly at the beginning, but later changed to an abbreviation? For example, changing the exploratory factor analysis to EFA. Revised after verification.

Introduction.

Lines 86-89, page 2: “It is not merely a simple 86 evaluation of knowledge about technology; it also grasps individual tendencies. It can help in providing education tailored to the characteristics of participants who use new technology.” This sentence should be placed at the end of the previous paragraph, which also explains the advantages of ATF. This sentence should be placed at the end of the previous paragraph, which also elaborates on ATI's advantages. In short, there should be a paragraph explaining the significance of the research.

Materials and Methods

2.1. Participants

Lines 94-99, page 3: This paragraph has repeated statements before and after. You can directly write down the inclusion criteria and exclusion criteria (which can include reasons).

Lines 100-109, page 3: You mentioned that each question requires 5 to 10 samples, how many questions does your questionnaire include? Correspondingly, what is the final required sample size? Whether the sample size collected in the study is sufficient should also be clearly written? The text is too messy, I didn't see your modifications clearly.

2.2. Measures

Lines 115-116, page 3: “In this study, it was .79.” One of the research purposes of this article is to explore its "reliability"? Why do you put the research results here.

Lines 117-124, page 3: Is this scale self-made or a mature scale cited? Its Cronbach's did you analyze it yourself or did you obtain it from existing literature? If obtained from literature, please insert the corresponding reference.

Conclusions

The conclusion section is still chaotic. Please elaborate on what the research has identified for your research purpose? What practical problems have been solved? What is the significance for practice? What is the impact on subsequent research?

In short, 1) you should provide me with a manuscript that does not contain any modification marks, otherwise it will be difficult for me to know exactly what you want to change. 2) The logic between sentences should also be strengthened. 3) Avoid redundant semantic expressions.

Thank you and my best,

Your reviewer

Minor editing of English language required

Author Response

Reviewer 2 Report (Previous Reviewer 2)

Tables and figures still lack more details in their descriptions. 

Make a sanity check of the language, but no larger issues were found. 

Round 2

Reviewer 1 Report (Previous Reviewer 1)

ID: healthcare-2457533

Title: Reliability and validity analysis of the Korean version of the Affinity for Technology Interaction scale

Thank you for providing a chance to review this manuscript.

Comment:

The author has made good modifications to the previous suggestions, and now in my opinion, there are no more suggestions.

Thank you and my best,

Your reviewer

Minor editing of English language required

This manuscript is a resubmission of an earlier submission. The following is a list of the peer review reports and author responses from that submission.

Round 1

Reviewer 1 Report

ID: healthcare-2276030

Title: Reliability and validity analysis of the Korean version of the Affinity for Technology Interaction scale

Thank you for providing a chance to review this manuscript.

Comment: Major Revision.

Detailed information:

Abstract

Page 1, Line 11-24: 1) The boundary between the methods, results and conclusions of this part is not clear, which leads to the unclear summary of the article in the summary. Please improve. 2) Please list the main data or values in the result section.

Introduction

Page 1-2, Line 28-59: The description of these paragraphs is to illustrate the social context, thereby eliciting the use and concept of ATI, which can be appropriately simplified.

Page 4, Line 77-84: Please indicate the reliability, validity, and usage of the original ATI scale.

Overall: There should be a progressive relationship between paragraphs, rather than random stacking. Please strengthen the logic of this part.

Materials and Methods

Participants, Line 92-95: 1) Your sample size is too small and I haven't seen your thinking about sample size. Please solve this problem. How are participants recruited? In addition, the Inclusion and exclusion criteria are too crude. Please be more careful. 2) Online surveys? How is quality control done?

Results

Table 4: Please indicate the full name of each scale below the table.

Overall:1) It is better to use "three-line table", that is, a table composed of three horizontal lines, to make the results more concise and clear. 2) Where are the results of Reliability?

Discussion

Page 7, Line 220-227: “Furthermore, research in the field of problem-solving (Robertson, 1996) and behavior or self-regulation (Carver, 2006) has long argued the existence of individual differences related to interaction styles. For example…” I don't particularly understand what this passage is trying to say.

Page 8, Line 244-255: There are many similar references in the introduction, and this paragraph could be simplified or deleted.

Overall: The logical order of the first four paragraphs is out of order, and it is recommended to reorganize. Discussions can be written in the order of the results sections.

Conclusions: Too verbose, please simplify.

In general, the biggest problem with your research is the insufficient sample size. Please provide sufficient evidence or supplement the sample size. In addition, some of the statements in the article are confusing, and the logic needs to be strengthened. Try modifying them.

Thank you and my best,

Your reviewer

Reviewer 2 Report

To enable a broader usage of the ATI scale is good, of course, and the researchers succeeded to involve 154 nurses in their study which is good.  It would have been interesting to not only translate the scale, but to actually deploy it as well. As the scale is very generic towards all types of technology, does the type matter? The interaction needed? In which situations? The main motivation for the study is that nurses are and will be even more exposted to tech in their work, but the implications of a new training scheme is hardly discussed (the consequenses of using the scale to "divide" the nurses into different training groups (as I see as a result)). Also, the motivation for doing the research comes very late in the paper (page 8). I also wonder why certain personal details were collected about the participants - like marital status and if they were religious or not - did this info impact the results? Also, I lack a more nuanced view regarding your results in light of your user data - as you had 154 nurses participating, could you find user groups/patterns in your collected data that we can learn from? Tables and figures need more detailed descriptions. There are some minor grammar issues in the text, but the researchers will be able to spot these.

Round 2

Reviewer 1 Report

ID: healthcare-2276030

Title: Reliability and validity analysis of the Korean version of the Affinity for Technology Interaction scale

Thank you for providing a chance to review this manuscript.

Comment: Minor Revision.

Detailed information:

Keywords

Line 24: The key words should include the word "scale". After all, the purpose of your research is to test the reliability and validity of a certain scale.

Introduction

Page 2, Line 57-60: The transition between ‘In addition, user coping and adaptation to a new system is associated with knowledge of the 58 system, degree of intimacy, general ability to interact with the technology system, individual disposition, and personality’ and the previous sentence is very stiff. Please modify to strengthen the logic.

Materials and Methods

Participants, Line 94-96: Although you have explained how to choose the sample size, how many questions do you have? What is the corresponding expected total sample size? Whether the sample size collected in the study is sufficient should also be clearly written.

Measures, Line 106: “The Cronbach’s of ATI in a previous study was .83–.92 (Franke et 105 al., 2019). In this study, it was .80.” Isn't that the purpose of the study? Why is it listed here.

Measures, Line 113-114: Where does the value of "Cronbach's " come from? Do you have any references? In addition, explanatory language can be added to indicate whether the reliability of the scale is good.

Results

Validity, Line 205-207: Please unify the writing format of the notes below the table. Other forms should also be unified.

      Reliability, Line 210-212: Where does the value of "Cronbach's " come from? Why is it different from the methods listed in the previous section.

Discussion

Discuss in the order in which the results are presented, and try to present key information.

Conclusions

What is the significance for future research or other aspects?

Thank you and my best,

Your reviewer
